# Development and Validation of a Mobile Application as an Adjuvant Treatment for People Diagnosed with Long COVID-19: Protocol for a Co-Creation Study of a Health Asset and an Analysis of Its Effectiveness and Cost-Effectiveness

**DOI:** 10.3390/ijerph20010462

**Published:** 2022-12-27

**Authors:** Mario Samper-Pardo, Sandra León-Herrera, Bárbara Oliván-Blázquez, Belén Benedé-Azagra, Rosa Magallón-Botaya, Isabel Gómez-Soria, Estela Calatayud, Alejandra Aguilar-Latorre, Fátima Méndez-López, Sara Pérez-Palomares, Ana Cobos-Rincón, Diana Valero-Errazu, Lucia Sagarra-Romero, Raquel Sánchez-Recio

**Affiliations:** 1Department of Medicine, University of Zaragoza, 50009 Zaragoza, Spain; 2Department of Psychology and Sociology, University of Zaragoza, 50009 Zaragoza, Spain; 3Institute for Health Research Aragon (IISAragon), 50009 Zaragoza, Spain; 4Department of Physiatry and Nursing, University of Zaragoza, 50009 Zaragoza, Spain; 5Aragones Group of Research in Primary Health Care (GAIAP), 50009 Zaragoza, Spain; 6Department of Nursing, University of La Rioja, 26004 Logroño, Spain; 7GAIAS Research Group, Faculty of Health Sciences, University San Jorge, 50830 Zaragoza, Spain; 8Department of Preventive Medicine and Public Health, University of Zaragoza, 50009 Zaragoza, Spain

**Keywords:** long COVID-19, APP, health asset, co-creation, effectiveness, cost-efficiency, primary health care

## Abstract

Objective: To analyse the overall effectiveness and cost-efficiency of a mobile application (APP) as a community health asset (HA) with recommendations and recovery exercises created bearing in mind the main symptoms presented by patients in order to improve their quality of life, as well as other secondary variables, such as the number and severity of ongoing symptoms, physical and cognitive functions, affective state, and sleep quality. Methods: The first step was to design and develop the technologic community resource, the APP, following the steps involved in the process of recommending health assets (RHA). After this, a protocol of a randomised clinical trial for analysing its effectiveness and cost-efficiency as a HA was developed. The participants will be assigned to: (1st) usual treatment by the primary care practitioner (TAU), as a control group; and (2nd) TAU + use of the APP as a HA and adjuvant treatment in their recovery + three motivational interviews (MI), as an interventional group. An evaluation will be carried out at baseline with further assessments three and six months following the end of the intervention. Discussion: Although research and care for these patients are still in their initial stages, it is necessary to equip patients and health care practitioners with tools to assist in their recovery. Furthermore, enhanced motivation can be achieved through telerehabilitation (TR).

## 1. Introduction

Coronavirus (COVID-19), caused by the severe acute respiratory syndrome SARS-CoV-2, has had a serious impact on the whole world and has triggered unprecedented health, social, and economic crises [1]. Most patients infected by SARS-CoV-2 recover in no more than a month depending on the severity of their symptoms [2,3] However, it is believed that irrespective of the severity of symptoms, around 20% of people continue to show or develop multisystemic symptoms after five weeks or more following the acute infection, while 10% continue to display symptoms after twelve weeks, despite receiving negative PCR test results and developing antibodies [4].

In the absence of a clear definition, this new symptomology had been referred to interchangeably as “Long COVID” or “Post-COVID” [5,6,7]. It is in this context that the National Institute for Health and Care Excellence (NICE) states that COVID-19 symptoms can last for 4–12 weeks and that those whose symptoms continue beyond this period and with no alternative diagnosis will be considered to have “Post-COVID syndrome” [8]. In October 2021, the World Health Organisation (WHO) released an official definition of the condition, referring to it as “Post-COVID condition” [9], a condition that occurs in individuals who were previously likely to have been infected (without diagnostic test confirmation) or confirmed to have been infected with SARS-CoV-2. 

The characteristic symptoms of this new condition are confusing, unspecific, and can be ongoing or fluctuate over time [10,11]. The most prevalent symptoms include chronic fatigue, breathlessness, fever, coughing, headaches, chest pain and/or a sore throat, muscle pain, general aches and pains, palpitations, tingling, diarrhoea, rashes, myalgia, brain fog and neurological symptoms, as well as symptoms of depression and anxiety stemming from extreme fatigue and other physical symptoms, among other causes [12,13,14]. There is a higher prevalence among women than men, with the latter making up 20% of diagnoses [15].

There are still many unknown factors (diagnosis, etiology, and disease phenotypes) that are grouped under the umbrella of long COVID-19, and the response to this illness is in its initial stages [13,16]. Therefore, it is necessary to promote research, especially in the field of primary health care (PHC) [17], as it is these health care practitioners who routinely treat long COVID-19 patients. To generate this evidence, a broader outlook should be adopted that does not focus on deficit care models, and patients themselves should be involved through co-production in health and the opportunities presented by citizen science [18]. In this way, citizens can actively contribute to the research by employing their own intellectual effort, knowledge, tools, and resources. 

In the case of health issues, it will allow for the analysis and implementation of more viable interventions based on patients’ experiences, utilising the knowledge of their strengths and challenges they face. They should be included and involved in care planning with a view to co-production in health with contextualised and salutogenic biopsychosocial care models. Consequently, the recognition of patients’ individual, family, and community health assets (HAs) would be valuable. Understanding how these general resilience resources operate and enable greater consistency, would, in turn, allow them to understand, manage, and find purpose in their condition [19]. This project, based on a qualitative study, has taken the insights (practical wisdom) of those who are suffering from ongoing symptoms having contracted COVID-19 into account. Therefore, the needs of this group in their stages of recovery and the possible responses from the health system and especially from PHC have been jointly considered. 

The recommendation of health assets (RHA) or social prescribing (SP) is a tool that can be used in PHC. It complements the biopsychosocial model and allows for the contextualisation of care through a multidisciplinary and intersectoral approach based on health determinants while granting individuals and communities the necessary means to improve and exercise greater control over their wellbeing [20,21,22,23]. HAs can be defined as any factor or resource which enhances the ability of individuals, communities, and populations to maintain and sustain health and wellbeing. Whether at an individual, family, or community level, the common denominator granted by these assets is the strengthening of individuals or groups’ ability to maintain or improve physical, mental, or social health and to deal with stressful situations. HAs could be seen as general resources to overcome inequality and as essential to strengthen knowledge and skills with regard to maintaining good health, boosting morale, and empowering individuals and communities. As a consequence, there would be less vulnerability and dependence on the healthcare system [24,25]. RHA creates formal mechanisms through HAs to provide certain individuals with non-clinical healthcare alternatives. Thus, within the framework of therapy, we can recommend pre-existing non-clinical community resources. 

To implement RHA, it is important to understand the current state of affairs. Following the COVID-19 pandemic, there have been changes made to the use of many clinical and non-clinical resources. Within this context and as an alternative solution to varying requirements, the pandemic has led to increased use of new technologies in diverse settings, including HAs, to ensure efficient communication between practitioners and patients [26,27]. However, this change has been in the works for around a decade, with the promotion of new technology and its increased use among the general population [28]. A clear example of this is the introduction of telerehabilitation (TR): the provision of rehabilitation services through electronic systems using information and communication technology. TR goes beyond hospital settings, helping to evaluate the effectiveness of rehabilitation in everyday activities [29]. Numerous recent studies looking at different diseases have demonstrated the efficiency of using a mobile application (APP) in the treatment of symptoms [30,31,32]. In addition, using TR for post-COVID patients has been proven to be effective, leading to improvements in dyspnoea, fatigue, and ability [33,34]. To sum up, RHA is a technique used to treat a wide variety of patients, including those diagnosed with long COVID-19. 

Medical rehabilitative treatment options for these patients are still proving to be limited, imprecise, and not entirely effective. However, there are no large-scale studies demonstrating the efficacy of pharmacological treatment and, as such, it is essential to continue working on the rehabilitation options offered to these patients [35]. In addition, scientific evidence suggests that early rehabilitation is essential to regain normal functionality [36,37]. Rehabilitation options similar to those offered to patients with chronic fatigue syndrome have always been indicated, that is, gradual and personalised physical and respiratory exercise therapies led by professionals [38,39]. In addition, as the development of the symptoms has been confirmed, it has become necessary to offer cognitive–behavioural therapy and attention from mental health professionals, given the negative impact of the disease itself on the patient’s quality of life [40,41]. Based on this evidence, it seems necessary to consider all these types of therapies in addition to ensuring healthy lifestyle habits, such as diet or sleep quality, to develop a comprehensive rehabilitation plan for long COVID-19 patients.

In light of this challenge, while co-creating the treatment and rehabilitation process, the need to create a community RHA for this patient group to be implemented at the earliest convenience was identified. This was due to the urgency stemming from the number and variety of symptoms presented by patients, as well as the uncertainty that still exists concerning certain pathophysiological aspects of Long COVID. Given the lack of existing resources for the group concerned, the need to design and develop an APP was raised, called ReCoVery. As far as we know, there are only two projects with a design and with purposes such as the ones presented in this study. However, these other two projects have not yet published their results since they are in the development phase of the clinical trial [42,43]. One of the projects is based on therapies such as those used for patients with fibromyalgia given the shared inability to objectify some symptoms. The two studies will serve as precedents for this project. However, it is considered necessary to design and implement a specific rehabilitation plan for this group of patients, which considers all the effects on their vital areas.

For the reasons outlined above, it would be a community resource, designed through a process of identification and co-creating content with Long COVID patients, who could later use the APP to support their recovery by means of TR. Hence, the objective of this study is to analyse the overall effectiveness and cost-efficiency of an APP as a community HA with recommendations and recovery exercises created bearing in mind the main symptoms presented by patients in order to improve their quality of life, as well as other secondary variables, such as the number and severity of ongoing symptoms, physical and cognitive functions, affective state, and sleep quality. A secondary objective is the analysis of personal factors, such as activation, self-efficacy, or health literacy as mediators of the effectiveness of the treatment.

## 2. Materials and Methods

### 2.1. Methodology to Design and Develop the Community Resource (ReCoVery APP)

This methodology is based on the steps involved in the RHA process.

#### 2.1.1. Start and Contextualisation

Firstly, the project was prepared and contextualised. The main objective was then established as offering rehabilitation through community resources to PHC patients diagnosed with long COVID-19 in the Spanish region of Aragón in order to improve their quality of life, as well as to alleviate their symptoms. To this end, a multidisciplinary team of general practitioners, nurses, psychologists, social workers, physiotherapists, and occupational therapists was created. To contextualise the environment and patient group, a qualitative study was carried out with semi-structured interviews and focus groups in order to identify their needs as well as the resources to be used during the treatment and, particularly, to delve deeper into the identification of community HAs during their recovery.

#### 2.1.2. Identification and Characterisation of Possible Community Activities 

Secondly, after detecting the patients’ lack of knowledge and difficulties due to their specific symptoms in the use or suitability of community resources, the multidisciplinary team was able to identify and characterise usable community activities. A list of activities carried out in the community was created, establishing certain criteria for the selection of activities in order to create a “map of HA for Long COVID patients”. After contacting interlocutors from each resource/activity, it was concluded by consensus that there were no sufficient rehabilitation community resources for this patient group according to the criteria established by the multidisciplinary team. Therefore, the need to create an ad hoc developed resource for this group that was also in line with the needs and demands established in the qualitative study, was considered. As a result, an APP with recovery content was designed and was treated as an HA in itself that, as requested by its users, needed to suit their needs.

#### 2.1.3. Building Community Connection

Thirdly, this phase was dedicated to the process of creation and development of the APP ReCoVery. The iterative method selected for the development of ReCoVery was human-centred design [44,45]. This technique looks to resolve specific problems based on the understanding of the needs and different perspectives of the users themselves as possible recipients of the intervention. In this way, it is possible to promote an adapted and personalised treatment, as well as to ensure adherence to it. For this reason, the initial design of ReCoVery was guided by symptoms, detected needs, and other information that was obtained through individual qualitative interviews and focus group discussions with patients diagnosed with long COVID-19. Subsequently, available scientific evidence was collected on health recommendations and recovery exercises for patients with Long COVID. Within the multidisciplinary team, different work subgroups were created to work on the design and creation of content for each rehabilitation area.

Prior to the start of the intervention, the content of the APP was periodically evaluated both by patients with Long COVID (future users) and by general practitioners, nurses, physiotherapists, occupational therapists, social workers, and psychologists to ensure its suitability and facilitate its adaptation and transfer following the combination between the SCRUM methodology [46] and the agile methodology, with the latter being optimal for software development [47]. In this case, the creation of the APP was distributed in various incremental and interactive development cycles (sprints). At the end of each sprint, the selected patients and the multidisciplinary team had to give their approval or suggest possible modifications that were consensually implemented.

Regarding the architecture of the APP, a native mobile APP with Java language was created through Android Studio, making it compatible with Android, one of the main platforms. Therefore, the APP can be run on all Android smart devices with a software version higher than 5.0. The design of a native APP was chosen, instead of a hybrid one, in order to obtain improved performance in the long term, as well as to be able to make use of the device’s own tools, such as notifications, thus allowing the APP to be kept up to date.

ReCoVery consists of six main modules, as detailed below, each one including a rehabilitation area for long COVID-19 patients, with various options and levels of adaptation being available to meet the specific needs of the patients, as well as self-report registration. In addition, the APP sends periodic reminder notifications, with configuration options.

Regarding the download and use, for new users this APP will not be publicly available, thus guaranteeing that it will only be downloaded onto the intended devices. The APP will be installed on the mobile devices of the participants during the first intervention session, where they will be provided with a username and the personal access password necessary to log in. In the event of having to reinstall the APP, due to a change in device or any other situation, it will be necessary to contact the research staff to make an appointment and carry out the installation process again.

ReCoVery will automatically record usage data, including logins, visits to each module and specific content, time of use, and any other interaction with the APP. In addition, each module consists of daily self-assessments where the self-perception of the users is reported. These data create statistical graphs regarding evolution in the different areas, allowing the user to see their evolution with a time perspective. The answers to these self-assessments will be stored in a SQL server database in the cloud hosted in Azure and will also be locally saved on the patient’s own device in a SQLite database. ReCoVery ensures the privacy and security of user data. In addition to carrying out the download process on-site, an authentication process must be completed for the installation by providing an identification code and a personal password. This information is only requested the first time the patient logs in, meaning that access to the APP will be remotely revoked in the event of loss, theft, or any reported improper use after contacting the research staff. 

With regard to field tests, users will be asked to report any bugs or crashes in the APP, as well as suggest possible modifications for future improvement. Brief individual qualitative interviews will also be conducted to further detail the ideal experience that the patients would prefer to have in relation to the functioning of ReCoVery. The research team will carry out new bibliographic research in order to obtain and incorporate new scientific evidence on the subject. These results were used to identify potential areas for improvement, and an updated version of ReCoVery (V.2.0) will be created.

#### 2.1.4. Recommendation of the APP to Patients

Fourthly, the established protocol for the recommendation of the APP was implemented. To this end, PHC general practitioners in Aragón were asked to identify potential patients who could then be offered the possibility to participate. Following this referral, the evaluating researcher contacted the patients to confirm or decline their participation. They were then invited to the research group’s headquarters for baseline tests, and the APP was recommended to them. This process continued until there was a sufficient sample size, as detailed below. Moreover, means of communication between practitioners and patients were established to monitor the intervention and resolve any arising problems.

#### 2.1.5. Evaluation and Revitalisation

Finally, evaluations could be carried out, in accordance with the previously established parameters, through the users’ contributions and following the quality of health promotion interventions, with the ultimate goal of offering this HA to the whole community. 

Figure 1 shows the flowchart of the APP’s design and development process.

### 2.2. APP’s Validation, Effectiveness and Cost-Efficiency as a HA

#### 2.2.1. Study Design

Randomised ecological clinical trial was represented in two parallel groups. The two interventions to which the patients will be assigned will be: (1st) usual treatment by the primary care practitioner (TAU—control group); and (2nd) TAU + use of the APP as a HA and adjuvant treatment in their recovery, plus three face-to-face motivational interview sessions to enhance adherence (intervention group).

#### 2.2.2. Study Population

The study population will be people with ongoing COVID symptoms who are aged 18 years or over and being treated by PHC. Exclusionary criteria are: diagnosis of a serious uncontrolled illness, which may interfere with the APP recommendations; significant risk of suicide; pregnancy and breastfeeding; participation in another clinical trial within the last six months; existing structured rehabilitative or psychotherapeutic treatment by health professionals and the presence of any medical, psychological or social problem that may significantly interfere with the patient’s participation in the study.

Patients will be recruited by PHC professionals. They will explain the nature of the study and invite patients to participate. Patients from the Association of Long COVID Patients in Aragón will also participate. Recruitment will happen consecutively until the sample size will be reached. 

#### 2.2.3. Sample Size

To calculate the sample size, we will use the data obtained in the study of Dalbosco-Salas et al. [34] because the intervention is similar to ours, and it is developed in PHC, despite being a clinical trial. To the best of our knowledge, to date there are no studies carried out with post-COVID-19 patients that evaluate a similar intervention with a clinical trial methodology. We will use the difference pre-post score of the short-form 36-item questionnaire (SF-36), the value of the possible highest standard deviation (SD), and a minimum expected difference of 19.3 points in the pre-post rating, which were all taken into account. Accepting a risk alpha of 0.05 and a power of 95% in a bilateral contrast, 70 subjects would be required. A maximum loss-to-follow-up rate of 10% has been estimated. The total sample required is 78 subjects, distributed between the two groups (control–intervention).

#### 2.2.4. Patient Inclusion

When the PHC practitioner identifies a potential participant, they will fill out a referral form indicating that the patient meets the criteria and then provide the patient with an information leaflet. The health care professional will inform the researcher once the patient has confirmed their consent. The evaluating researcher will then contact the participant and confirm whether or not they will be included in the study based on the inclusion and exclusion criteria. 

#### 2.2.5. Randomisation, Allocation and Masking of Study Groups

Once baseline data have been collected, the participants will be randomised. An independent statistician will perform the individual randomisation using a computer-generated random number sequence (blinded sequence). The randomisation will be carried out using a list of patients (Figure 2). Given the nature of the interventions, participants will not be blind to their allocation. A researcher will call them to explain their assigned intervention and will request that participants do not inform other researchers of their allocation.

#### 2.2.6. Intervention

1st Recommended treatment (TAU) prescribed by health care professionals in accordance with NICE Guidelines and CAMFiC’s Manifestaciones persistentes de la COVID-19: Guía de práctica clínica [Ongoing manifestations of COVID-19: Clinical Practice Guidelines] and its subsequent updates.

2nd TAU plus the use of the APP as a community resource + three face-to-face motivational interviewing sessions to increase adherence. 

The content of the APP is based on scientific evidence that aims to alleviate the symptoms of patients diagnosed with long COVID-19 [48,49,50,51,52]. Depending on their health status, the participants will find all the advises useful, or only those that respond to their health needs based on the persistent symptoms they present will be helpful. However, it is emphasized that, if they cannot carry out any of the recommendations or exercises, or it causes them any type of discomfort, they should stop doing them immediately. The information provided in the APP has been divided into six work areas: (1) recommendations for following the Mediterranean diet; (2) recommendations for improving the sleep quality; (3) physical exercise recommendations with visual aids (i.e., charts); (4) respiratory physiotherapy exercises supported by video tutorials; (5) cognitive stimulation exercises with different levels of difficulty and (6) use of community resources. There is an additional section on behavioral activation and evidence-based documentation on managing long COVID-19. In all these sections, self-assessments and records are provided to record their activity so that patients can graphically observe their evolution over time as they continue to use the APP.

The content and operation of each of each section is detailed below:Diet

According to the Clinical Guide for Long COVID patient care of the Spanish Society of General and Family Physicians (SEMG) [49], eating habits should be taken into account to supply possible nutritional deficiencies. This APP aims to provide healthy eating pattern recommendations based mainly on the adherence to the classic Mediterranean diet. Specifically, the intake of vitamin D, vitamin B12, B complex, folic acid, and omega-3 fatty acids is recommended, according to some previous studies [53,54]. These nutrients are involved in the proper functioning of the metabolism, as well as in the immune system, so they could be beneficial for long COVID-19 patients. For the monitoring and evaluation of patients, this APP will offer a weekly evaluation based on 8 questions that will measure the patient’s adherence to the Mediterranean diet, showing an evolutionary chart of it.

2.Sleep hygiene

The quality of sleep and rest are fundamental components to feel physically and psychologically well. The amount of sleep needed varies by person and age, but most adults need between 7 and 8 h of sleep each night [55,56,57]. Given the great negative impact that has been found in various studies regarding the quality of sleep of long COVID-19 patients, it has been considered necessary to offer sleep guidelines to sleep well and improve rest [58,59]. However, in case of serious sleep problems, it is recommended to consult with a medical professional. This APP performs a daily evaluation of the self-perception that people have about their rest through 3 questions, presenting a history chart of it. 

3.Physical exercise

It has been shown that physical exercise can have benefits in multiple pathologies with which the long COVID-19 syndrome shares similarities both in its symptoms and possible pathogenic mechanisms. In this specific syndrome, the data are still insufficient, but the latest recommendations emphasize the need for personalized and symptom-adjusted physical activity to reduce long COVID-19 symptoms and promote recovery in these people [52]. Within this APP basic physical, exercises will be offered. This content was developed by a physiotherapist, and both the physical exercised and the instructions provided to the APP’s user patients were based on the recommendations proposed by different guides about the management of this disease or other pathologies with similar symptoms available in the current scientific evidence [50,51]. 

4.Respiratory physiotherapy

For the treatment of persistent respiratory symptoms, such as dyspnea, the different existing guidelines about the management of this disease recommend breathing exercises, that is, respiratory physiotherapy [49,51,60]. That is why a physiotherapist developed the respiratory exercises proposed in the content of this APP, based on the guidelines mentioned above. Similar to physical activity, this type of exercise must be graded and personalized to the characteristics of each patient, something that is stated in the instructions prior to carrying out these exercises.

5.Cognitive exercises

The most common cognitive impairment profile in long COVID-19 involves impaired executive function or planning, difficulty sustaining attention, decreased processing speed, deficits in short-term memory, abstraction and orientation, and even language impairment and anomie [61]. That is why an occupational therapist specialized in neurology developed three levels (i.e., easy, medium difficulty, difficult) of cognitive stimulation exercises aimed at working on all the cognitive skills that may be affected in the adult population with long COVID-19. In addition, a series of complementary voluntary activities were developed so that patients can stimulate all of the aforementioned within their activities of daily living (e.g., remembering a shopping list to stimulate memory).

6.Community resources: socialization and emotional well-being

This section, based on the effectiveness of RHA [20,21,22,23], mentioned in the introduction section, aims to improve the physical and emotional wellbeing of the participants through integration in the community, as well as socialization with peers. To achieve a good quality of life, it is not only essential to take care of our body through good habits, but also necessary to know how to live in the community (i.e., that our relationships with others are healthy). Therefore, the purpose of this section is to promote the participation in the process of local development through the use of different services, associations, or cultural activities, as well as groups affected by the same pathology.

In addition to the use of the APP, the intervention group will attend three face-to-face sessions held over three consecutive weeks with the aim of encouraging adherence to the APP, based on a motivational interview (MI). The first and second sessions will be individual, and the third will be a group session. All of them will be conducted by two study researchers with training to standardise the intervention, as well as with specific training, to carry out these sessions in accordance with Miller and Rollnick’s guidelines [62]. MIs will be conducted for 20–30 min, in alignment with the health belief model, which identifies the link between treatment adherence and health behavior [63,64,65].

This way of working follows Miller and Rollnick’s [62] definition of MI, which is “A collaborative, goal-oriented style of communication with particular attention to the language of change. It is designed to strengthen personal motivation for and commitment to a specific goal by eliciting and exploring the person’s own reasons for change within an atmosphere of acceptance and compassion”. Thus, MI is an interviewing method centred around the patient, which aims to evoke behavioural change by helping patients to understand and solve their problems [66]. Additionally, in the first session, the APP will be installed, and the participant will be taught how to manage and use it. The second session will be focused on sharing experiences of the first week of using the APP, addressing technological and content concerns, and setting goals they want to achieve through its use. The third session will be in groups (around 10–15 participants in each group) and will last for approximately one hour. In this last session, the benefits of using the APP and other community HAs as an adjuvant treatment will be discussed. 

#### 2.2.7. Variables and Instruments

An evaluation will be carried out at baseline with further assessments three and six months following the end of the intervention. The assessors administering the instruments will be blind to the type of treatment given to the patients. Patients will be contacted by phone at six weeks to monitor their adherence to APP recommendations and exercises in the intervention group and important changes in their health (e.g., reinfection) or treatment in both the control and intervention groups. 

The main variable will be quality of life, assessed by the SF-36 Questionnaire [67], which measures eight dimensions of health: physical function, physical role, aches and pains, general health, vitality, social function, emotional role, and mental health. In addition, it incorporates a declared health progress item. The eight dimensions define two main components of health: physical summary component and mental summary component, where scores above or below 50 indicate better or worse health status, respectively, than the mean of the reference population. The items are scored on Likert-type scales ranging from 1 to 3.5 or 6 depending on the type of item. The eight scales are scored from 0 to 100, with higher scores indicating better health status. The official Spanish version of the questionnaire [68] will be used.

Secondary results: The following variables will be collected

-Socio-demographic variables: gender, age, civil status, education, household, and occupation. Roles will also be collected using the Spanish version of the Role Checklist, whose test–retest reliability, measured by weighted Kappa, is 0.74 [69,70], an inventory divided into two parts. The first part evaluates the presence of the ten main roles of people’s life over time. Individuals should indicate whether they have performed each of the roles in the past (any time up to the week immediately preceding the assessment), whether they are currently being performed (on the day the checklist is completed and during the seven days prior), and if they plan to perform them in the future (any time from the following day). It is possible to mark more than one time for each role. The second part measures the value that the individual attributes to each role (“Not at all valuable”, “Somewhat valuable”, or “Very valuable”). People should mark the value they consider for each of ten roles, even if they have never played them or do not plan to do so in the future [71].-Clinical variables: clinical history, contraction of COVID-19, timeline of developing Long COVID, number of residual symptoms, and their severity measured via an analogue visual scale [72], days taken on sick-leave. Residual symptoms include: gastrointestinal symptoms, loss of smell, loss of taste, blurred vision, eye problems (increased dioptre, dry eyes, conjunctivitis), tiredness or fatigue, cough, fever (over 38 °C), low-grade fever (37–38 °C), chills or shivering without fever, bruising, myalgia, headaches, sore throat, dyspnoea, chronic fatigue, dizziness, tachycardia, orthostatic hypotension, joint pain, chest pain, back pain (cervical, dorsal or lumbar), neurological symptoms (tingling, spasms, etc.), memory loss, confusion or brain fog, short attention and concentration span, loss of libido or erectile dysfunction, altered menstrual cycle, urinary symptoms (infections, overactive bladder), hair loss, and other symptoms that can be considered residual [73,74].-Cognitive variables:(a)To assess the presence of cognitive impairment, the official Spanish version of the Montreal Cognitive Assessment (MoCA) [75,76,77] will be used, which is a test with adequate internal consistency (Cronbach’ alpha of 0.76) that assesses six cognitive domains (memory; visuospatial ability; executive function; attention, concentration or working memory; language; and temporo-spatial orientation). It is out of a total score of 30 points, and a correction of one point can be made in the case of subjects with fewer than 12 years of schooling. The cut-off point for the detection of mild cognitive impairment in its original version is 26. This test has been used to assess cognitive impairment of people with long COVID-19 [78,79].(b)The Symbol Digit Modalities Test (SMDT) will also be used to detect dysfunction related to divided attention, visual tracking, perceptual, and motor speed and memory, both in children and adults, and with a test-retest reliability of between 0.84 and 0.93 in a sample of healthy adults. It consists of converting a series of 120 symbols of different shapes into the numbers that correspond to each one following the key provided. This must be conducted consecutively and as quickly as possible within 90 s after completing a 10-digit practical test. The total score is obtained by counting the number of correct substitutions completed out of a maximum score of 110. A score below 33 is considered a clear indicator of some type of cognitive disorder [80,81].(c)To measure short-term memory impairment, the Spanish version of the Memory Impairment Screen (MIS) will be used. This brief test assesses the existence of memory disorders using free recall (without clues) or selective recall (with semantic clues) of four words. In dementia screening, it presents adequate interobserver (0.85) and test–retest (0.81) reliability. Two points are awarded per word obtained by free recall and one point per word recalled with the help of semantic clues. The total scores range from zero to eight, with a score of four or less indicating possible cognitive impairment [82,83].(d)To assess whether verbal fluency is affected, the Semantic Verbal Fluency Test (Animals) (test-retest reliability of 0.68) will be used, which consists of counting the number of correct words reproduced in 1 min within the category ‘Animals’. Normally, a person without impairment will be able to reproduce about 16 words in 1 min [84,85].-Functional physical variables:(a)Cardiorespiratory capacity will be measured by a 6 min walk test (6MWT) [86]. It is a functional cardiorespiratory test that measures the maximum distance a subject can walk for 6 min. The test measures and records baseline and post-test heart rate, oxygen saturation (SpO2), and dyspnea according to the Borg scale [87]. The 6MWT walk had good test-retest reliability (88 < R < 94). We will use the most recent official Spanish version [88].(b)Leg strength and endurance will be measured by Sit to Stand Test [89]. We will use 30-s Sit to Stand Tests, which are used specifically to test for respiratory diseases [90]. The test evaluates endurance at a high power, speed, or velocity in terms of muscular or strength endurance by recording the number of times a person can stand up and sit down completely in the space of 30 s. The 30-s chair stand has good test-retest reliability (84 < R < 92). We will use the 30 s Sit to Stand Test that has been translated in Spanish and used for COVID-19 patients [91].-Affective state through the Hospital Anxiety and Depression Scale (HADS) questionnaire [92]. The HADS is a scale based on self-report that was developed to detect depression and anxiety disorders in medical patients in primary care settings. The HADS includes 14 items that assess symptoms of anxiety and depression (HADS-A and HADS-D, respectively), with each item corresponding to a 4-point (zero to three) scale, with scores ranging from 0 to 21 for symptoms of both anxiety and depression, with higher scores indicating more severe symptoms. The HADS has been translated into a number of languages, including Spanish [93], to facilitate its use in international trials [94].-Sleep quality through the Insomnia Severity Index (ISI). The ISI [95] works through self-reporting and measures a patient’s perception of nocturnal and diurnal symptoms of insomnia: difficulties initiating sleep, staying asleep, early morning awakening, satisfaction with current sleep pattern, interference with daily functioning, noticeability of impairment attributed to sleep deprivation, and degree of distress or concern caused by sleep deprivation. This scale has seven items, with each answer ranging from zero to four, and an overall score ranging from 0 to 28, with a higher score indicating a higher severity of insomnia. The Spanish version of the ISI [96] shows an adequate internal consistency (Cronbach alpha = 0.82). It has also been used in other studies of people with long COVID-19 [97].

Social support will be measured by the Medical Outcomes Study Social Support Survey (MOS-SS) [98]. It is a self-report instrument consisting of four subscales (emotional/informational, tangible, affectionate, and positive social interaction) and an overall functional social support index. It has good reliability (Cronbach’s alpha ≥ 0.91) and is quite stable over time. It has 19 items, as well as a 5-point Likert Scale. Higher scores indicate more support. We will use the official Spanish version [99]. Community social support will also be assessed using the Perceived Community Support Questionnaire (PCSQ) [100], which provides information on social resources as perceived by the members of the community. It consists of 25 questions assessing four aspects: community integration, community participation, social support from informal systems, and social support from formal systems. It is a Likert-type questionnaire with a scale of one to five. The authors have identified the reliability of the different scales to range between 0.75 and 0.88, which was assessed using the Cronbach’s alpha coefficient [100,101]. In previous research, it has been found to provide an adequate assessment of community experiences of both adults and young people [102]. 

-Physical activity will be measured using the International Physical Activity Questionnaire-Short Form (IPAQ-SF) [103]. It assesses the levels of habitual physical activity over the preceding seven days. It has seven items and records activity at four levels of intensity: vigorous-intensity activity and moderate-intensity activity (walking and sitting). We will use the official Spanish version [104]. IPAQ-SF has sufficient validity for the measurement of total and vigorous physical activity and poor validity for moderate activity and good reliability [105].-Adherence to a Mediterranean diet will be measured using the 14-item Mediterranean Diet Adherence Screener (MEDAS), encouraging compliance to a Mediterranean diet [106]. It includes items on food consumption and intake habits. The total score ranges from 0 to 14, with a higher score indicating greater adherence to the Mediterranean diet [107].-Personal constructs. The personal factors relating to behaviour that will be collected are the following:(a)Self-efficacy will be measured using the Self-Efficacy Scale-12 [108]. The original scale consisted of 17 items that are scored on a 5-point Likert scale. Woodruff and Cashman [109] obtained a factor structure, based on the original 17-item scale, that represented the three aspects underlying the scale, i.e., willingness to initiate behavior, `Initiative’, willingness to expend effort in completing the behavior, `Effort’, and persistence in the face of adversity, `Persistence’. Five items were excluded because of low item-rest correlations and ambiguous wording, resulting in a 12-item version of the scale (GSES-12). This scale has 3 factors: Initiative (willingness to initiate behavior), Effort (willingness to make an effort to complete the behavior), and Persistence (persevering to complete the task in the face of adversity). Internal consistency of the original scale was 0.64 for initiative, 0.63 for effort, and 0.64 for persistence. The total scale obtained a Cronbach’s Alpha coefficient of 0.69 [110].(b)Patient activation in their own health will be measured using the Patient Activation Measure (PAM) questionnaire with regard to the management of their health [111]. It evaluates the patient’s perceived knowledge, skills, and confidence to engage in self-management activities through 13 items with a Likert Scale from one (strongly disagree) to four (strongly agree). The resulting score ranges between 13 and 52. Higher scores indicate higher levels of activation. There is only an official Spanish version for chronically ill patients. It has an item separation index for the parameters of 6.64 and a reliability of 0.98 [112].(c)Health literacy will be measured using the Health Literacy Europe Questionnaire (HLS-EUQ16) [113]. Health literacy is defined as the knowledge of the population, their motivation, and their individual ability to understand and make decisions related to the promotion and maintenance of their health. The questionnaire consists of 16 items, scored between 1 (very easy) and 4 (very difficult). The score of each subject was obtained as the sum of the scores of the 16 items. The final score can be transformed into a dichotomous response: very difficult and difficult = 0, as well as easy and very easy = 1. Higher scores indicate worse health literacy. It presents a high consistency (Cronbach’s alpha of 0.982) in the official Spanish version [114].-For the analysis of the cost-efficiency, the Client Service Receipt Inventory will be used [115], collecting information on the entire range of services and support used by study participants. It retrospectively collects data on the use of services over the preceding six months (e.g., rates of use of individual services, mean intensity of service use, rates of accommodation use over time). We will use the official Spanish version [116].

#### 2.2.8. Statistical Analysis 

Analysis of the outcomes at baseline consist of the following measures. First, descriptive analysis of all of the variables (frequencies for categorical variables; means and standard deviation for continuous variables) will be carried out. A univariate analysis (one-way ANOVA and chi-square) will be used to examine the data and test whether there are baseline differences between groups after randomisation.

Statistical analyses will be chosen based on the sub-sample size (parametric or non-parametric tests). Data collection and statistical analyses will be performed using Excel software, SPSS software (version 25.0) [117], and the R statistical software environment (version 3.6.2) [118]. 

Clinical effectiveness analysis: The report of the results will follow a pre-specified plan based on CONSORT guidelines [119] in order to compare the two groups. Initially, a descriptive comparison (proportions, means, or medians) will be carried out between groups for prognostic variables in order to establish their baseline comparability after randomisation. To analyse the clinical effectiveness, a repeated-measure linear regression will be conducted, including all evaluations over time. For this purpose, the main variable, SF-36 score, will be used as a continuous variable. The models will include adjustments for the baseline value of the SF-36 and for any other variable that would show differences in the baseline measurement. Possible group per time interactions will be examined using linear regression. Similar analyses will be carried out using the secondary outcomes (number and severity of persistent symptoms, Montreal Cognitive Assessment, Sit-to-Stand Test, and HAD test). To counteract the problem of multiple comparisons we will use Bonferroni correction.

Cost-efficiency and cost-utility analysis are as follows. The effectiveness of the interventions will be estimated using the difference between the SF-36 score at the baseline and at the three and six-month follow-up, and the utility will be estimated using QALYs at the three and six-month follow-up. QALYs will be calculated based on these scores using the Spanish EQ-5D tariffs [120]. Along with the number and severity of the ongoing symptoms, scores will also be used as an outcome for the analysis. Cost-efficiency will be explored through the calculation of incremental cost-efficiency ratios (ICERs) for the intervention group using the TAU group as the control. ICER is defined as the ratio between incremental costs and incremental effectiveness. In the same way, cost utility will be explored through the calculation of incremental cost-utility ratios (ICURs), which are defined as the ratio between incremental costs and incremental utilities measured on QALYs. QALYs gained in each evaluation are approximated by using the area under-the-curve technique [121]. Total costs will be calculated by adding direct and indirect costs. Direct costs will be calculated by adding the costs derived from the medication and the use of health services and clinical tests. The medication costs will be calculated by determining the price per milligram during the study period according to the Vademecum during the last year of the study, including value-added tax (VAT). The total cost of drug treatment will be calculated by multiplying the price per milligram by the daily dose in milligrams and the number of days the treatment is received. Costs derived from the use of health services will be calculated considering the data from the Oblikue database [122]. Indirect costs will be calculated based on the number of days taken on sick-leave and multiplying them by the Spanish minimum daily wage during the study period 2019–2020. We assume that data will be missing at random (MAR). Only patients with both cost and relevant outcome data at three and six-month follow-up will be included in the cost-efficiency and cost-utility analyses. Notwithstanding, sensitivity analysis imputing missing three and six-month data will test the robustness of the cost-efficiency and cost-utility results. The imputations will be performed using the package “mice” [123], freely available in cran-R [118].

Variables collected, instruments, and measures are shown in Table 1. 

#### 2.2.9. Ethical Considerations

Ethical approval was granted by the Clinical Research Ethics Committee of Aragón (PI21/139 and PI21/454). The procedures carried out for the creation of this work complied with the ethical standards of the previously mentioned committee and with the 1975 Declaration of Helsinki. All of the subjects will sign a comprehensive consent form, their data will be anonymised, and will only be used for the purposes of the study. Participants and health care professionals will be informed of the results. Patients of the TAU group will be invited to use the APP at the end of the study. The ethics committee will be notified of any protocol modifications. 

## 3. Results

The initial development of the ReCoVery APP has been completed at the design and architecture level. The therapeutic content (videos, information, and other types of media) has been uploaded to the APP, allowing for a fully functional version. Regarding the APP’s validation, effectiveness, and cost-efficiency study, we are now completing the recruitment phase. To date, 60 patients have participated in the study, being assessed and randomized.

## 4. Discussion

COVID-19 has severely impacted the global population, especially long COVID-19 patients. It is estimated that the worldwide prevalence of this syndrome approaches 10% [126]. The treatment and care of these patients have become a social responsibility, which should be addressed by future research. Additionally, given the health and symptoms experienced by those suffering from long COVID-19, TR is considered a potentially useful tool for the rehabilitation of these patients [127]. This is a global health issue and, therefore, a shared global response is necessary.

To the best of our knowledge, no randomized clinical trials have been performed using TR on Long COVID patients. However, effective studies based on respiratory, physical, or cognitive TR with other groups of patients support the potential viability of this study. As for respiratory TR, two bibliographic reviews of controlled trials of TR for the provision of pulmonary rehabilitation have concluded that TR may be as effective as face-to-face pulmonary/respiratory rehabilitation in individuals suffering from chronic respiratory diseases [128,129]. However, a randomized controlled trial by Cox et al. (2022) concluded that respiratory TR may not be as effective as pulmonary rehabilitation, but it is safe for patients and would have clinically significant benefits [130]. More similar to long COVID-19 patients, a pilot study by González-Gerez et al. (2021) verified that respiratory TR is effective, safe, and feasible for COVID-19 patients having mild-moderate symptoms in the initial stage [131]. Furthermore, the study by Liu et al. (2020) associated respiratory rehabilitation with an improved quality of life in patients infected with COVID-19 [132]. Similarly, the benefits of TR based on physical activity may become comparable with conventional face-to-face rehabilitation approaches while also producing less interference in the patient’s daily life [133]. In this line, the study by Nambi et al. (2021) identified significant improvements in the physical component of the low-intensity aerobic activity group in COVID-19 patients [134]. Additionally, a pilot study by Abodonya et al. (2021) verified a significant improvement in the quality of life of COVID-19 patients who underwent a TR intervention based on physical activity [135]. In addition, prior evidence verifies that TR based on physical exercises may improve the quality of life of groups of patients, such as those suffering from multiple sclerosis [136,137]. Finally, a systematic review has verified that cognitive TR produces benefits in various cognitive domains, such as verbal fluency and executive functions [138]. 

Various studies, however, have revealed that early rehabilitation interventions are essential for post-COVID patients to avoid progressive deterioration before the symptoms become chronic [139]. Therefore, in addition to physical, respiratory, and cognitive interventions, this study has also included healthy lifestyle habits (adherence to the Mediterranean diet, sleep hygiene, and RHA), following the existing clinical guidelines for long COVID-19 patients in order to improve the quality of life of these patients.

This project has strengths and limitations. The strengths of this study include the design of a qualitative needs assessment and the co-creation of a community-based TR resource that can be widely used. It will provide a wealth of information on the interplay between the quality of life of people with long COVID, ongoing symptoms, and personal factors on health behaviour. Study limitations include the possible attrition of participants due to reinfections leading to health deterioration or seeking other health care resources during the course of the study. However, possible reasons for attrition and other issues will be recorded.

## 5. Conclusions

Although the research and care of these patients is still in its early stages, there is a need to equip patients and healthcare professionals with tools to help them in their recovery. In this line, it is worth highlighting the role of PHC and community nursing, turning out to be professional mediators between general practitioners and the community. Through this complementary work, it is intended to access comprehensive health care, obtain the acquisition of habits, capacities, and behaviors that promote self-care. 

## Figures and Tables

**Figure 1 ijerph-20-00462-f001:**
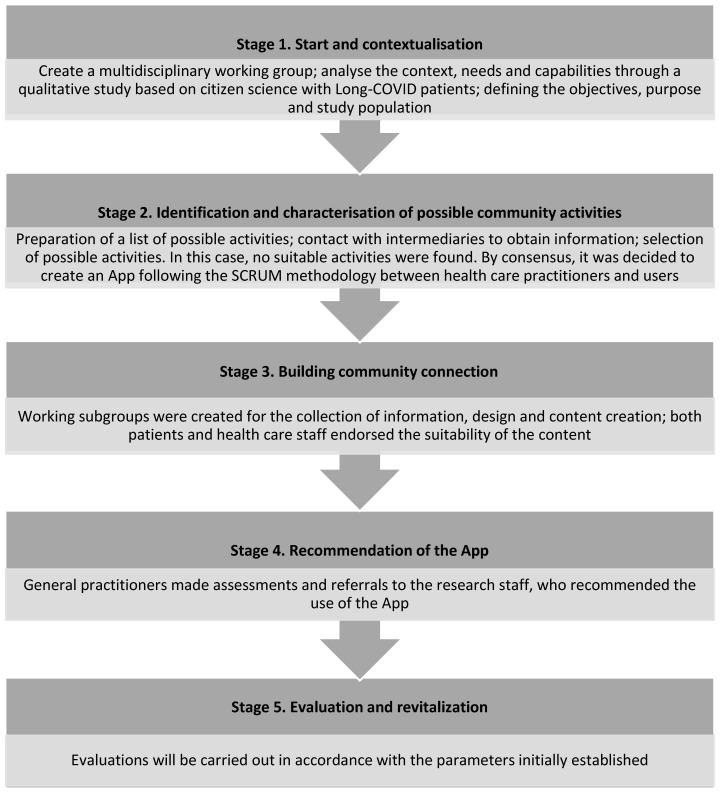
Design process and recommendation of a rehabilitation community resource (APP).

**Figure 2 ijerph-20-00462-f002:**
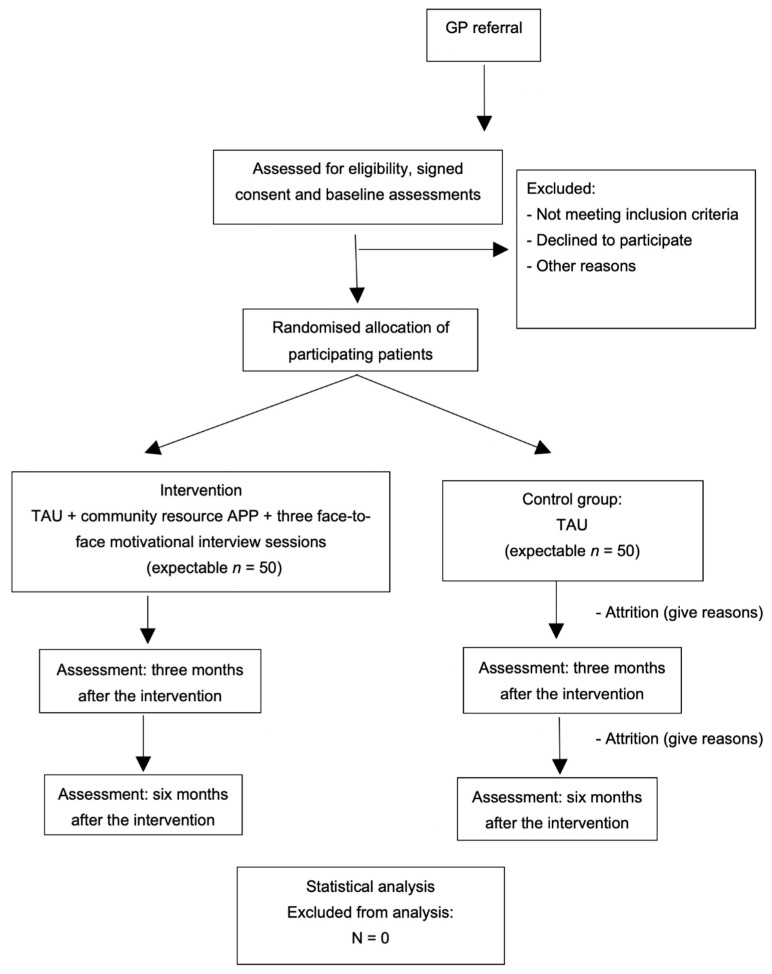
Flowchart of the study: randomisation, sampling, and monitoring of patients.

**Table 1 ijerph-20-00462-t001:** Study variables.

Instruments	Assessment Areas
Gender, ages, civil status, education, household, occupation. Role Checklist [71]	Socio-demographic variables
Clinical history, contraction of COVID-19, timeline of developing Long COVID, number of residual symptoms and their severity (EVA), days taken on sick-leave [73,74]	Clinical variables
SF-36 [68,124]	Quality of life
Montreal Cognitive Assessment [75,76,77] The Symbol Digit Modalities Test (SMDT) [80,81] Memory Impairment Screen (MIS) [82,83] Semantic Verbal Fluency Test (Animals) [84,85]	Cognitive variables
6 min walk test (6MWT) [86] Sit to Stand Test 30 sg [89]	Functional physical variables
HADS [92]	Affective state
Insomnia Severity Index (ISI) [95]	Sleep Quality
Medical Outcomes Study Social Support Survey (MOS-SS) [98] Perceived Community Support Questionnaire [125]	Social Support
International Physical Activity Questionnaire-Short Form (IPAQ-SF) [103]	Physical Activity
14-item Mediterranean Diet Adherence Screener (MEDAS) [106]	Adherence to a Mediterranean diet
Self-Efficacy Scale [108] Patient Activation Measure Questionnaire (PAM) [111] Health Literacy Europe Questionnaire (HLS-EUQ16) [113]	Personal constructs
Client Service Receipt Inventory [115]	Social and health services used

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
