# Peer review of "Development and Validation of a Mobile Application as an Adjuvant Treatment for People Diagnosed with Long COVID-19: Protocol for a Co-Creation Study of a Health Asset and an Analysis of Its Effectiveness and Cost-Effectiveness"

_ijerph, 2022, doi:10.3390/ijerph20010462_

Round 1
Reviewer 1 Report (New Reviewer)
Dear authors,
Your work presents a very interesting perspective on the field. The background and the context are well presented.
However, more valuable comments can be entailed to present the outcomes.
Also, the discussion section needs to be improved. It is much too short, being extremely necessary to present your results in the context. For a worthy section according to this paper, it must refer to past studies and how your research complements them.
I wish you good work with your work!
Author Response
Thank you very much for your comments.
Your considerations have been taken into account and I would like to comment on the following aspects with their respective modifications:
- As it is a protocol, it is not possible to present numerical results yet. The information included in results section is like other publications. However, a paragraph has been added expressing the intention of the researchers involved to disseminate the future results.
“The initial development of the ReCoVery APP has been completed at the design and architecture level. The therapeutic content (videos, information and other types of media) has been uploaded to the APP, allowing for a fully functional version. Regarding the APP’s validation, effectiveness and cost-efficiency study, we are now completing the recruitment phase. At the moment, 60 patients have been included in the study, assessed and randomized.
- In accordance with your request, it has been decided to expand the discussion section. To this end, the effectiveness that telerehabilitation can have in patients with other chronic diseases has been reinforced, since there are no large-scale experiences of telerehabilitation with Long COVID patients. It is considered that these modifications have contributed to contextualize the feasibility of this is study.
“COVID-19 has severely impacted everyone and especially Long COVID patients. It is estimated that the worldwide prevalence of this syndrome is around 10% (131). The treatment and care of these patients has become a social responsibility, which should be addressed by future research. In addition, due to the health situation and the symptoms presented by Long COVID patients, it is estimated that TR can be a tool to rehabilitate these patients (132). It is a global health problem and therefore there must be a shared global response. To the best of our knowledge, randomized clinical trials based on TR in Long COVID patients have not been performed. However, effective studies based on respiratory, physical or cognitive TR with other groups of patients could support the viability of the present study. Regarding respiratory TR, two bibliographic reviews of controlled trials of TR for the provision of pulmonary rehabilitation conclude that TR may be as effective as face-to-face pulmonary respiratory rehabilitation in people with chronic respiratory diseases (133,134). However, the randomized controlled trial by Cox et al., (2022), concludes that respiratory TR may not be as effective as pulmonary rehabilitation, but it is safe for patients and would achieve clinically significant benefits (135). Closer to Long COVID patients, the pilot study by González-Gerez et al., (2021) verified that respiratory RT is effective, safe and feasible for COVID-19 patients with mild-moderate symptoms in the initial stage (136). Furthermore, the study by Liu et al. (2020) associated respiratory rehabilitation with an improvement in the quality of life in patients infected with COVID-19 (137). In the same way, the benefits of physical activity TR can become comparable with conventional face-to-face rehabilitation approaches, in addition to producing less interference in the daily life of patients (138). In this line, the study by Nambi et al. (2021) identified significant improvements in the physical component of the low-intensity aerobic activity group in COVID-19 patients (139). Also, the pilot study by Abodonya et al. (2021), verified a significant improvement in the quality of life of COVID-19 patients who underwent a physical activity TR intervention (140). In addition, previous evidence verifies that TR based on physical exercises can improve the quality of life of groups of patients such as those with multiple sclerosis (141,142). Finally, a systematic review reinforces that cognitive RT produces benefits in various cognitive domains such as verbal fluency or executive functions (143). However, several studies ensure that early rehabilitation interventions are essential for post-COVID patients to avoid progressive deterioration until symptoms become chronic (144). For this reason, in addition to physical, respiratory and cognitive interventions, this study has decided to offer healthy lifestyle habits (adherence to the Mediterranean diet, sleep hygiene or RHA), following the existing clinical guidelines for Long COVID patients, with the aim of improving the quality of life of these patients”.
Once again, thank you for your contributions.

Reviewer 2 Report (New Reviewer)
I would like to congratulate the team of authors who took the initiative to develop a computer application to manage post-COVID symptoms. It is a wonderful idea that will help patients.
I have a few comments about this article.
I propose the collective of authors modify in the title of the word APP with the long form of the word to ensure a good understanding of the message. The exclusive use of capital letters within a word suggests an abbreviation, which in the present case does not apply.
Also, I suggest you review Figure 2. „Flowchart of the study: randomisation, sampling and monitoring of patients”, where one word is not visible in the box on the round 328-329.
Congratulations on the hard work done.
Thanks for the opportunity to review this paper.
Author Response
Thank you very much for your comments.
Your considerations have been taken into account and I would like to comment on the following aspects with their respective modifications:
- The word "APP" has been modified in the title by "mobile application", as suggested in order to facilitate understanding.
- Figure 2 had suffered alterations due to the format, so it has been reconfigured.
Once again, thank you for your comments.

This manuscript is a resubmission of an earlier submission. The following is a list of the peer review reports and author responses from that submission.
Round 1
Reviewer 1 Report
This paper aims at analyzing the overall effectiveness and cost-efficiency of an APP as a community health asset with recommendations and recovery exercises created bearing in mind the main symptoms presented by patients in order to improve their quality of life. This paper first designed and developed the technological community resource (APP) following the steps involved in the process of recommending community assets. It then developed a protocol of a randomised clinical trial for analysing its effectiveness and cost-efficiency as a health asset. This paper is timely, interesting and well-written. I have two comments as follows.
1. This study proposed a new health technology for patients. But, what is the innovation of this technology compared with the existing similar technologies?
2. This study adopted a longitudinal between-subject experiment to test the effectiveness and cost-efficiency of the developed APP. However, the experimental design was not clear and convincing. Moreover, the data analysis and results are not well-presented.
Author Response
Thank you very much for your comments and appraisals. First of all, the English language has been reviewed along the manuscript.
- This study proposed a new health technology for patients. But, what is the innovation of this technology compared with the existing similar technologies?
The introduction section has been extended in order to provide more information about the innovation of the technology comparing to existing similar technologies.
The following has been added:
Medical rehabilitative treatment options for these patients are still proving to be limited, imprecise and not entirely effective. However, there are no large-scale studies demonstrating the efficacy of pharmacological treatment and, as such, it is essential to continue working on the rehabilitation options offered to these patients (35). In addition, scientific evidence suggests that early rehabilitation is essential to regain normal functionality (36,37). Rehabilitation options similar to those offered to patients with chronic fatigue syndrome have always been indicated, that is, gradual and personalised physical and respiratory exercise therapies led by professionals (38,39). In addition, as the development of the symptoms has been confirmed, it has become necessary to offer cognitive-behavioural therapy and attention from mental health professionals, given the negative impact of the disease itself on the patient’s quality of life (40,41). Based on this evidence, it seems necessary to consider all these types of therapies in addition to ensuring healthy lifestyle habits, such as diet or sleep quality, to develop a comprehensive rehabilitation plan for long-term patients with COVID.
In light of this challenge, while co-creating the treatment and rehabilitation process, the need to create a community RHA for this patient group to be implemented at the earliest convenience was identified. This was due to the urgency stemming from the number and variety of symptoms presented by patients as well as the uncertainty that still exists concerning certain pathophysiological aspects of long COVID. Given the lack of existing resources for the group concerned, the need to design and develop an application (APP) was raised, called ReCoVery. As far as we know, there are only two projects with a design and with purposes like the ones presented in this study. However, these other two projects have not yet published their results since they are in the development phase of the clinical trial (42,43). One of the projects is based on therapies like those used for patients with fibromyalgia given the shared inability to objectify some symptoms. The two studies will serve as precedents for this project. However, it is considered necessary to design and implement a specific rehabilitation plan for this group of patients, which considers all the effects on their vital areas.
- Gorbalenya AE, Baker SC, Baric RS, de Groot RJ, Drosten C, Gulyaeva AA, et al. The species Severe acute respiratory syndrome-related coronavirus: classifying 2019-nCoV and naming it SARS-CoV-2. Vol. 5, Nature Microbiology. Nature Research; 2020. p. 536–44.
- Ali RMM, Ghonimy MBI. Post-COVID-19 pneumonia lung fibrosis: a worrisome sequelae in surviving patients. Egyptian Journal of Radiology and Nuclear Medicine. 2021 Dec 13;52(1):101.
- Stam H, Stucki G, Bickenbach J. Covid-19 and Post Intensive Care Syndrome: A Call for Action. J Rehabil Med. 2020;52(4):jrm00044.
- Puchner B, SAHANIC S, KIRCHMAIR R, PIZZINI A, SONNWEBER B, WÖLL E, et al. Beneficial effects of multi-disciplinary rehabilitation in postacute COVID-19: an observational cohort study. Eur J Phys Rehabil Med. 2021 May;57(2).
- Torjesen I. NICE backtracks on graded exercise therapy and CBT in draft revision to CFS guidance. BMJ. 2020 Nov 10;m4356.
- Naidu SB, Shah AJ, Saigal A, Smith C, Brill SE, Goldring J, et al. The high mental health burden of “Long COVID” and its association with on-going physical and respiratory symptoms in all adults discharged from hospital. European Respiratory Journal. 2021 Jun;57(6):2004364.
- Titze-de-Almeida R, da Cunha TR, dos Santos Silva LD, Ferreira CS, Silva CP, Ribeiro AP, et al. Persistent, new-onset symptoms and mental health complaints in Long COVID in a Brazilian cohort of non-hospitalized patients. BMC Infect Dis. 2022 Dec 8;22(1):133.
- Blanchard M, Backhaus L, Ming Azevedo P, Hügle T. An mHealth App for Fibromyalgia-like Post-COVID-19 Syndrome: Protocol for the Analysis of User Experience and Clinical Data. JMIR Res Protoc. 2022 Feb 4;11(2):e32193.
- Murray E, Goodfellow H, Bindman J, Blandford A, Bradbury K, Chaudhry T, et al. Development, deployment and evaluation of digitally enabled, remote, supported rehabilitation for people with long COVID-19 (Living With COVID-19 Recovery): protocol for a mixed-methods study. BMJ Open. 2022 Feb 7;12(2):e057408.
- This study adopted a longitudinal between-subject experiment to test the effectiveness and cost-efficiency of the developed APP. However, the experimental design was not clear and convincing. Moreover, the data analysis and results are not well-presented.
The methodology section has been improved in order to clarify the manuscript and the procedures.
This manuscript is a research protocol, so there are not still available results. For this reason there are not analyses data and results.
Before submitting the manuscript, on behalf of the authors, I asked the editorial office of IJERPH and the director of the special issue if the research protocol manuscripts were accepted, and I obtained a positive answer.
However, A result section has been included in order to clarify this point and the timing. The following has been added:
- Results
The initial development of the ReCoVery APP has been completed at the design and architecture level. The therapeutic content (videos, information and other types of media) has been uploaded to the application, obtaining a fully functional version. Waiting to start the clinical trial.
Reviewer 2 Report
The theme of the paper closely follows the situation. The APP connects patients, professionals, and community environment elements to implementing RHA. The research does not focus on deficit care models. And through non-clinical community resources, it helps with recovery therapy for patients with long COVID. This research has specific theoretical and practical significance for healthcare management. This paper introduces the research background, experimental design, and data analysis methods in detail to help understand the research's starting point, rationality, and extension. However, some defects still can not be ignored and affect the integrity of the research report.
First, the objective in the Abstract is not the same as in the Introduction. The Introduction explains the research topic and design background in detail, but the current research status is briefly described. What is the progress of research on coping with long COVID, and what is the quality of research on similar platforms to the experiment to achieve health recovery? Third, the experimental time and data are not shown and explained in the Materials and Methods. Most importantly, no results are presented. The discussion needs to be based on the results and interpreted in the context of previous studies. Moreover, the Conclusions are not qualified.
Author Response
Thank you very much for your comments. First of all, the English language has been reviewed along the manuscript.
First, the objective in the Abstract is not the same as in the Introduction.
The objective in the abstract has been modified in order to be the same as in the introduction.
The Introduction explains the research topic and design background in detail, but the current research status is briefly described. What is the progress of research on coping with long COVID, and what is the quality of research on similar platforms to the experiment to achieve health recovery?
The introduction section has been extended in order to provide more information about the innovation of the technology comparing to existing similar technologies.
The following has been added:
Medical rehabilitative treatment options for these patients are still proving to be limited, imprecise and not entirely effective. However, there are no large-scale studies demonstrating the efficacy of pharmacological treatment and, as such, it is essential to continue working on the rehabilitation options offered to these patients (35). In addition, scientific evidence suggests that early rehabilitation is essential to regain normal functionality (36,37). Rehabilitation options similar to those offered to patients with chronic fatigue syndrome have always been indicated, that is, gradual and personalised physical and respiratory exercise therapies led by professionals (38,39). In addition, as the development of the symptoms has been confirmed, it has become necessary to offer cognitive-behavioural therapy and attention from mental health professionals, given the negative impact of the disease itself on the patient’s quality of life (40,41). Based on this evidence, it seems necessary to consider all these types of therapies in addition to ensuring healthy lifestyle habits, such as diet or sleep quality, to develop a comprehensive rehabilitation plan for long-term patients with COVID.
In light of this challenge, while co-creating the treatment and rehabilitation process, the need to create a community RHA for this patient group to be implemented at the earliest convenience was identified. This was due to the urgency stemming from the number and variety of symptoms presented by patients as well as the uncertainty that still exists concerning certain pathophysiological aspects of long COVID. Given the lack of existing resources for the group concerned, the need to design and develop an application (APP) was raised, called ReCoVery. As far as we know, there are only two projects with a design and with purposes like the ones presented in this study. However, these other two projects have not yet published their results since they are in the development phase of the clinical trial (42,43). One of the projects is based on therapies like those used for patients with fibromyalgia given the shared inability to objectify some symptoms. The two studies will serve as precedents for this project. However, it is considered necessary to design and implement a specific rehabilitation plan for this group of patients, which considers all the effects on their vital areas.
- Gorbalenya AE, Baker SC, Baric RS, de Groot RJ, Drosten C, Gulyaeva AA, et al. The species Severe acute respiratory syndrome-related coronavirus: classifying 2019-nCoV and naming it SARS-CoV-2. Vol. 5, Nature Microbiology. Nature Research; 2020. p. 536–44.
- Ali RMM, Ghonimy MBI. Post-COVID-19 pneumonia lung fibrosis: a worrisome sequelae in surviving patients. Egyptian Journal of Radiology and Nuclear Medicine. 2021 Dec 13;52(1):101.
- Stam H, Stucki G, Bickenbach J. Covid-19 and Post Intensive Care Syndrome: A Call for Action. J Rehabil Med. 2020;52(4):jrm00044.
- Puchner B, SAHANIC S, KIRCHMAIR R, PIZZINI A, SONNWEBER B, WÖLL E, et al. Beneficial effects of multi-disciplinary rehabilitation in postacute COVID-19: an observational cohort study. Eur J Phys Rehabil Med. 2021 May;57(2).
- Torjesen I. NICE backtracks on graded exercise therapy and CBT in draft revision to CFS guidance. BMJ. 2020 Nov 10;m4356.
- Naidu SB, Shah AJ, Saigal A, Smith C, Brill SE, Goldring J, et al. The high mental health burden of “Long COVID” and its association with on-going physical and respiratory symptoms in all adults discharged from hospital. European Respiratory Journal. 2021 Jun;57(6):2004364.
- Titze-de-Almeida R, da Cunha TR, dos Santos Silva LD, Ferreira CS, Silva CP, Ribeiro AP, et al. Persistent, new-onset symptoms and mental health complaints in Long COVID in a Brazilian cohort of non-hospitalized patients. BMC Infect Dis. 2022 Dec 8;22(1):133.
- Blanchard M, Backhaus L, Ming Azevedo P, Hügle T. An mHealth App for Fibromyalgia-like Post-COVID-19 Syndrome: Protocol for the Analysis of User Experience and Clinical Data. JMIR Res Protoc. 2022 Feb 4;11(2):e32193.
- Murray E, Goodfellow H, Bindman J, Blandford A, Bradbury K, Chaudhry T, et al. Development, deployment and evaluation of digitally enabled, remote, supported rehabilitation for people with long COVID-19 (Living With COVID-19 Recovery): protocol for a mixed-methods study. BMJ Open. 2022 Feb 7;12(2):e057408.
Third, the experimental time and data are not shown and explained in the Materials and Methods.
Experimental time data has been explained in results section, adding the following:
- Results
The initial development of the ReCoVery APP has been completed at the design and architecture level. The therapeutic content (videos, information and other types of media) has been uploaded to the application, obtaining a fully functional version. Waiting to start the clinical trial.
Some changes have been made along the manuscript in order to clarify the material an methods section.
Most importantly, no results are presented. The discussion needs to be based on the results and interpreted in the context of previous studies. Moreover, the Conclusions are not qualified.
This manuscript is a research protocol, so there are not still available results. For this reason there are not analyzed data and results, neither a discussion section interpreting the results nor conclusions supported by the results.
Before submitting the manuscript, on behalf of the authors, I asked the editorial office of IJERPH and the director of the special issue if the research protocol manuscripts were accepted, and I obtained a positive answer.
However, A result section has been included in order to clarify this point and the timing. The following has been added:
- Results
The initial development of the ReCoVery APP has been completed at the design and architecture level. The therapeutic content (videos, information and other types of media) has been uploaded to the application, obtaining a fully functional version. Waiting to start the clinical trial.